# Geometry Matching for Multi-Embodiment Grasping

**Maria Attarian**[1,2], **Muhammad Adil Asif**[2], **Jingzhou Liu**[2], **Ruthrash Hari**[2], **Animesh Garg**[2,3],
**Igor Gilitschenski**[2], **Jonathan Tompson**[1]

[1]Google DeepMind, [2]University of Toronto, [3]Georgia Institute of Technology

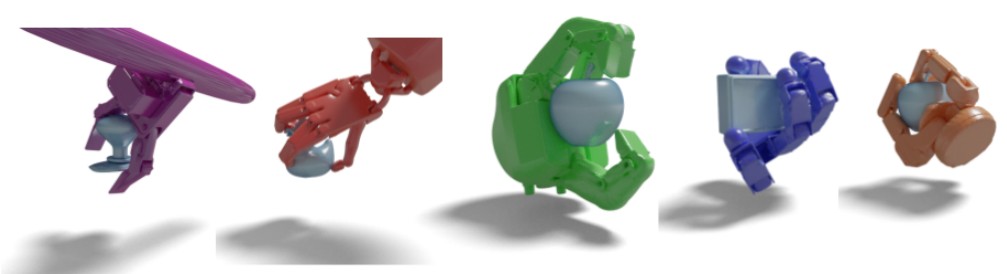

Figure 1: **GeoMatch**: Our method enables multi-embodiment grasping by conditioning the grasp selection on end-effector and object geometry.

**Abstract:** Many existing learning-based grasping approaches concentrate on a single embodiment, provide limited generalization to higher DoF end-effectors and cannot capture a diverse set of grasp modes. We tackle the problem of grasping using multiple embodiments by learning rich geometric representations for both objects and end-effectors using Graph Neural Networks. Our novel method - *GeoMatch* - applies supervised learning on grasping data from multiple embodiments, learning end-to-end contact point likelihood maps as well as conditional autoregressive predictions of grasps keypoint-by-keypoint. We compare our method against baselines that support multiple embodiments. Our approach performs better across three end-effectors, while also producing diverse grasps. Examples, including real robot demos, can be found at `geo-match.github.io`.

**Keywords:** Multi-Embodiment, Dexterous Grasping, Graph Neural Networks

## 1 Introduction

Dexterous grasping remains an open and important problem for robotics manipulation. Many tasks where robots are involved boil down to some form of interaction with objects in their environment. This requires grasping objects with all kinds of different geometries. In addition, the large variety of robot and end-effector types necessitates that grasping should also be achievable with new and arbitrary end-effector geometries. However, the cross-embodiment gap between grippers does not permit simply applying grasping policies from one end-effector to another, while domain adaptation i.e. "translating" actions from one embodiment to another, is also not straightforward. In comparison, humans are extremely versatile: they can adapt the way they grasp objects based on what they know about object geometry even if the object is new to them, and they can do this in multiple ways.

There has been much research in grasping thus far, with many works focusing on one embodiment at a time [1, 2, 3, 4, 5, 6, 7, 8, 9, 10, 11] and fewer looking at the multi-embodiment problem [12, 13, 14]. Methods are divided between hand-agnostic or hand-aware, and experiment with different representations for grasping, such as contact maps [12], contact points [13] or even root pose and

---

Correspondence emails: jmattarian@google.com, adil.asif@mail.utoronto.ca

7th Conference on Robot Learning (CoRL 2023), Atlanta, USA.

z-offset [14]. Existing multi-embodiment approaches either require explicit representation of joint limits that becomes exponentially harder in higher DoF end-effectors, or expect heavy manual work to adapt to new end-effectors, or showcase mixed rates of success across different embodiments with some gripper/hand morphologies performing significantly better than others.

Drawing inspiration from how humans adapt their grasps easily and successfully based on priors they have learned about 3D geometry of both objects in space and their own hands, we propose endowing robotic agents with a similar sense of geometry via a generalized geometry embedding that is able to represent both objects and end-effectors. This embedding can be used to predict grasps that demonstrate stability, diversity, generalizability, and robustness. More specifically, we propose to rely on Graph Neural Networks (GNN) to encode meaningful geometry representations and use them to predict keypoint contacts on the object surface in an autoregressive manner.

In summary, our contributions are as follows:

a) We propose formulating robot grasp learning as a geometry matching problem through learning correspondences between geometry features, where both, end-effector and object geometries, are encoded in rich embedding spaces.
b) To solve the above problem, we introduce GeoMatch, a novel method for learning expressive geometric embeddings and keypoint contacts.
c) We demonstrate that our method is competitive against baselines without any extra requirements to support higher DoF end-effectors and while also showcasing high performance across multiple embodiments. Finally, we provide demos of our method on a real Franka Emika with an Allegro hand.

## 2   Related Work

**Dexterous Grasping.** Many dexterous grasping works do not look into multi-embodiment and instead focus on diversity of objects using a single end-effector. Many grasping methods support 2-finger parallel grippers [2, 3, 4, 5, 6, 7, 8, 15] with several others looking into high-DoF dexterous manipulation [9, 10, 1, 16, 17, 18]. Some work has also been conducted towards multi-embodiment grasping. Several of those address the problem from the differentiable simulation grasp synthesis point of view [19, 11, 20]. GenDexGrasp [12] advocate for hand-agnostic contact maps generated by a trained cVAE [21] with a newly introduced align distance, and optimize matching of end-effectors against produced contact maps via a specialized Adam optimization method. This is the most recent work to our knowledge, attempting to tackle multi-embodiment grasping without significant preprocessing to support higher DoF grippers. In contrast, we choose to operate on hand-specific contact maps as we are interested in learning both object and embodiment geometry conditioned grasps, and empirically found our method to perform more evenly well multi-embodiments.

Intuitively, our work is closest to UniGrasp [13]. UniGrasp operates on object and end-effector point clouds to extract features and ultimately output contact points which are then fed into an Inverse Kinematics (IK) solver, similarly to us. Their encoder of choice is PointNet++ and the contact prediction is done through a Point Set Selection Network (PSSN) [22]. Their proposed architecture adds one stage per finger, which means supporting more than 3 finger grippers requires manually adding another stage. As a result, adapting the method to more than 2-finger and 3-finger grippers requires significant work. Moreover, the method requires explicit representation of boundary configurations through combination of minimum and maximum values for each joint angle which explodes exponentially on higher DoF end-effectors. In contrast, we rely on learned geometry features to identify viable configurations as opposed to explicitly encoding them through joint limit representation. Our method only requires a small number of user-selected keypoints per end-effector, same number for all end-effectors. This disentangles the dependency between number of fingers and applicability of our method. Similarly to UniGrasp, EfficientGrasp [23] also uses PointNet++ and a PSSN model for contact point prediction. TAX-Pose [24] is another recent work that shares some high level concepts. Instead of encoding the end-effector, authors consider tasks involving objects that interact with each other in a particular way, and what the relative pose of interacting objects should be for the task to be considered successful. They proceed with encoder objects or object parts using DGCNN and learn a cross-attention model that predicts relative poses of objects that accomplish a task. AdaGrasp [14] uses 3D Convolutional Neural Networks to learn a scoring function for possible generated grasps, and finally executes the best one.

Finally, many of the methods mentioned, rely on deterministic solvers which can result in decreased diversity of generated grasps. While we also rely on a deterministic solver, we address this issue by leveraging the scoring we obtain by the learned full unnormalized distribution of contacts. The score is used to select a first keypoint that will guide the remaining contact point prediction. This permits higher diversity without having to sample a large number of grasps.

**Graph Neural Networks.** Graph Neural Networks were first introduced by Scarselli et al. [25] as a proposed framework to operate on structured graph data. Since then, many advancements have been made towards extending their capabilities and expressivity [26]. Specifically in the grasping literature, there have been multiple applications of GNN. E.g. Huang et al. [27] propose learning a GNN to predict 3D stress and deformation fields based on using the finite element method for grasp simulations. The use of GNNs for end-effector parameterization has been proposed before in [28] where tactile sensor data is fed into a GNN to represent the end-effector as part of grasp stability prediction. Lou et al. [29] leverage GNN to represent the spacial relation between objects in a scene and suggest optimal 6-DoF grasping poses. Unlike previous methods, we propose applying GNN as a general geometry representation for any rigid body, both objects and end-effectors jointly. For the purposes of this work, we leverage the GNN implementation by [30] due to the readily available and easily adaptable code base.

**Geometry-Aware Grasping.** Several works have emphasized the role of geometry in the grasping problem. Some works study geometry-aware grasping under the light of shape completion [31, 32]. Yan et al. [33] encodes RGBD input via generative 3D shape modeling and 3D reconstruction. Subsequently, based on this learned geometry-aware representation grasping outcomes are predicted with solutions coming out of an analysis-by-synthesis optimization. In the same vein, Van et al. [34] proposed leveraging learned 3D reconstruction as a means of understanding geometry, and further rely on this for grasp success classification as an auxiliary objective function for grasp optimization and boundary condition checking. Bohg et al. [35] introduced a classifier trained on labeled images to predict grasps via shape context. Finally, Jiang et al. [6] propose to learn grasp affordances and 3D reconstruction as an auxiliary task, using implicit functions. Unlike these works, we suggest looking at geometry itself directly from 3D as a feature representation without imposing any 3D reconstruction constraints.

# 3 Method

In this work, we aim to learn robust and performant grasping prediction from the geometries of both the object $\mathcal{G}_O$ and the end-effector $\mathcal{G}_G$. Our approach models the contact between pre-defined keypoints on the end-effector and contact points on the object surface. That is, we aim to learn a model $c_1, \ldots, c_N = \text{GeoMatch}\left(\mathcal{G}_O, \mathcal{G}_G, (k_i)_{i=0}^N\right)$, where $c_i \in \mathbb{R}^3$ are the predicted contact points on the object surface each of which corresponds to one of $N$ pre-defined keypoints $k_i \in \mathbb{R}^3$ on the end-effector.

**Method Overview.** We represent object and end-effector geometries using graphs $\mathcal{G}_O = (\mathcal{V}_O, \mathcal{E}_O)$, and $\mathcal{G}_G = (\mathcal{V}_G, \mathcal{E}_G)$, which can easily be generated from object point clouds and a URDF description respectively (Sec. 3.1). This formulation allows us to utilize GNNs to learn geometry-aware embeddings which are subsequently fed into an autoregressive grasp generation module (Sec. 3.2). The losses are used to supervise the graph embeddings in addition to supervising the output of the autoregressive module (Sec. 3.3). These losses require likelihood maps that can directly be obtained from the dataset (Sec. 3.4). At test time we sample the first keypoint randomly and execute the grasp using inverse kinematics (Sec. 3.5).

## 3.1 Object and End-effector Representations

To create the graphs, we generate a point cloud representing each end-effector in canonical pose. As a result, we have point cloud representations of both, the objects and the end-effectors. Those can directly be converted into graphs (described below). For the end-effector, we use the resulting graphs to manually select the canonical keypoints.

**End-effector representation.** We set the end-effector to a canonical rest pose $q_{\text{rest}}$. We chose the rest pose to be a vector with all joint angles in middle range of their respective joint limits, zero root

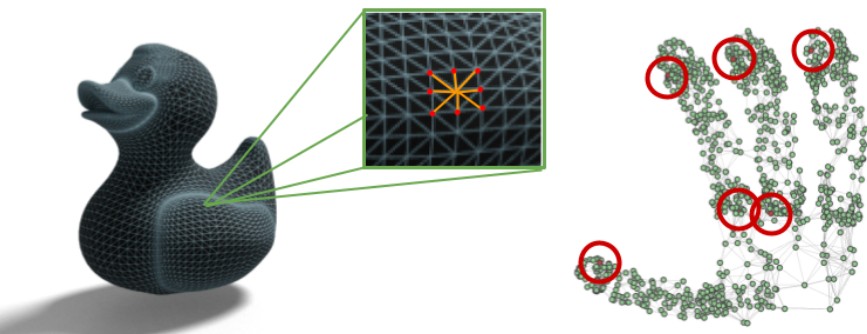

Figure 2: **Object and end-effector inputs.** Objects are initially represented as regularly sampled point clouds which are converted into a graph for further processing. End-effector geometries are given as meshes and converted to coarser graphs by randomly sampling points from the mesh as an intermediate step. User-selected keypoints are highlighted in red.

translation, and identity root rotation. In this pose we convert the end-effector mesh into a point cloud by sampling $S_G = 1000$ points from the mesh surface.

**Graph Creation.** The vertices of the graphs are obtained by simply considering each point in the point cloud a vertex. Similarly to the end-effectors, the number of points per object is the same for each object, in this work it is $S_O = 2048$. The edges are meant to capture local surface geometry and are created by connecting each point to its $K$ closest neighbours, a hyper-parameter that depends on point cloud density and object structure (we empirically chose $K = 8$).

**Canonical Contact Points.** The canonical contact points $k_i \in \mathcal{V}_G$ ($i \in \{1 \ldots N\}$) on the surface of each end-effector are selected visually. This needs only to be done once for each end-effector in a dataset ensuring that the selected keypoints have good coverage of each gripper with respect to its morphology and its grasping behavior. Please note that the order in which keypoints are selected is a hyper-parameter and might impact model performance. In our case, keypoint order was chosen to be as semantically consistent as much as possible across multiple embodiments (e.g. fingertips are ordered left to right, etc).

**Dataset.** For the purposes of this work, we use a subset of the MultiDex dataset introduced by [12] and used by them to train the CMap-VAE model of their approach. The dataset is comprised of 5 end-effectors - one 2-finger, two 3-finger, one 4-finger, and one 5-finger. 58 common objects from YCB [36] and ContactDB [37] are used. The dataset finally contains 50,802 diverse grasps over the set of hands and objects, each represented by an object name, an end-effector name, and the end-effector pre-grasp pose in the form:

$$q_{\text{rest}} = (t, R, \theta_0, ..., \theta_{N-1})_{\text{rest}}, \tag{1}$$

where $t \in \mathbb{R}^3$ is the root translation, $R \in \mathbb{R}^6$ is the root rotation in continuous 6D representation as introduced in [38], and $\theta_0, \ldots, \theta_{N-1}$ are the joint angles.

## 3.2 Architecture

Ideally, we would like to compute the full joint distribution of vertex-to-vertex contact, i.e. $P(v_o, v_g)$ for all object & end-effector vertices $v_o \in \mathcal{V}_O, v_g \in \mathcal{V}_G$. This is typically intractable. We reduce the complexity by only focusing on $N$ landmark keypoints $k_0, \ldots, k_N$ on the end-effector. We, thus, try to approximate $P(v_o, c_0, ..., c_N)$ where $c_i, i \in [0, N)$ are the vertices on the object where the $k_i$ keypoint makes contact, through learning a set of factorizations by applying the Bayes rule resulting in

$$P(v_o, c_0, ..., c_N) = \Pi_{i=1}^N P_{\mathcal{M}_i}(v_o, c_i | c_0, ..., c_{i-1}) = \Pi_{i=1}^N P_{\mathcal{M}_i}(v_o, c_i | \mathbf{c}_{<i})), \tag{2}$$

where $v_o, c_i \in \mathcal{V}_O$, $(k_0, ..., k_N) \subset \mathcal{V}_G$, and $P_{\mathcal{M}_i}(v_o, c_i | c_0, ..., c_{i-1})$ are the factorized marginals to be learned in an autoregressive manner.

We seek an architecture that can embed local geometry information well. From the architecture choices that demonstrate such properties, we chose Graph Neural Networks (GNN) [30] to create

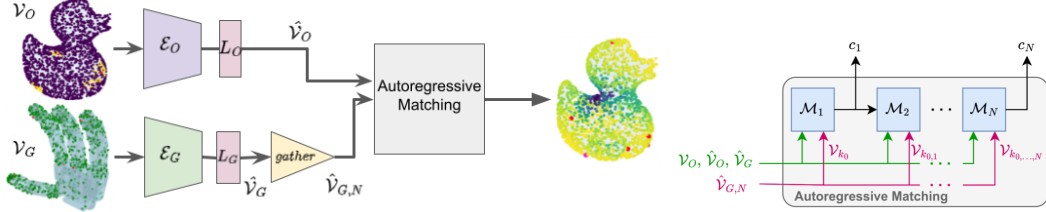

(a) Full overview of GeoMatch.                    (b) Autoregressive modules.

Figure 3: **GeoMatch architecture.** The object and gripper graphs are passed through the two encoders followed by linear layers. The gripper keypoint embeddings are gathered and are passed as input along with the object embeddings in the autoregressive modules.

object and end-effector embeddings. We, subsequently, gather the embeddings corresponding to the keypoints on the end-effector resulting in

$$\hat{\mathcal{V}}_O = \mathcal{E}_O(\mathcal{G}_O), \qquad \hat{\mathcal{V}}_G = \mathcal{E}_G(\mathcal{G}_G), \qquad \hat{\mathcal{V}}_{G,N} = \text{Gather}\left(\hat{\mathcal{V}}_G, (k_i)_{i=0}^N\right).$$

The gathered embeddings are passed into our autoregressive matching module jointly with the full object and end-effector embeddings, i.e.

$$c_0, \dots, c_N = \text{AutoregressiveMatching}(\hat{\mathcal{V}}_O, \hat{\mathcal{V}}_{G,N}).$$

**Geometry Processing.** For our experiments, we used the Graph Convolutional Networks (GCN) implementation by Kipf et al. [30] with 3 hidden layers of size 256 and 512 output embedding dimension, one for objects and one for end-effectors. The subsequent linear projections for each encoder $L_O$ and $L_G$ respectively, were of size 64 without bias.

**Autoregressive Matching.** The downprojected geometric embeddings are passed into an autoregressive matching module, consisting of 5 layers, $\mathcal{M}_i$. Each layer is responsible for predicting the index of the object vertex $c_n$ where keypoint $k_n$ makes contact, given outputs of previous keypoints $k_0, \dots, k_{n-1}$. This is done by concatenating the full geometric embeddings with vertex-keypoint distances. These are used as input to a vertex classifier that selects the resulting contact points $c_n$ on the object. Details are given in appendix A.1.

### 3.3   Losses

Our full training objective contains two components. One focusing on the geometric embeddings and another on the predicted contact points resulting in the overall loss

$$\mathcal{L}_{\text{total}} = \alpha \cdot \mathcal{L}_{P_{F_{0,\dots,N}}} + \beta \cdot \mathcal{L}_{P_{M_{0,\dots,N}}} \tag{3}$$

with each component having equal weight in our experiments ($\alpha = \beta = 0.5$).

**Geometric Embedding Loss $\mathcal{L}_{P_{F_{0,\dots,N}}}$.** The unnormalized likelihood map of object-keypoint contacts intuitively represents a score that a given object vertex is in contact with a given gripper keypoint and is given by

$$P_{F_i}(v_o, c_i) = \mathcal{E}_O(v_o) \cdot \mathcal{E}_G(v_g)[k_i]. \tag{4}$$

This is optimized against the dot product of the hand-specific object contact map $C_O(v_o, k_i)$ which will be explained further below, via a binary cross-entropy loss

$$\mathcal{L}_{P_{F_{0,\dots,N}}} = \Sigma_{i=1}^N \text{BCE}_{\lambda_a}(P_{F_i}(v_o, c_i), C_O(v_o, k_i)), \tag{5}$$

where $\lambda_a$ is the positive weight used to address the class imbalance (we use $\lambda_a = 500$).

**Predicted Contact Loss $\mathcal{L}_{P_{M_{0,\dots,N}}}$.** This loss models the joint distribution of contacts via a set of factorizations $P_{M_i}(v_o, c_i | c_0, \dots, c_{i-1}) \forall i \in [0, N)$ each of which are represented by the outputs of the autoregressive layers in our network. The outputs are optimized against the ground truth binary contact map label of the $i$-th gripper keypoint, contributing to a second binary cross-entropy loss term

$$\mathcal{L}_{P_{M_{0,\dots,N}}} = \Sigma_{i=1}^N \text{BCE}_{\lambda_b}(P_{M_i}(v_o, c_i | c_0, \dots, c_{i-1}), C_O(v_o, k_i)), \tag{6}$$

where, similarly, $\lambda_b$ is the positive weight hyperparameter used to address the class imbalance (we use $\lambda_b = 200$). A visual representation of the autoregressive layers can be seen in Fig. 3b. Note that for $i = 0$, $P_{F_0}(v_o, c_0)$ constitutes the first marginal for $k_0$ and thus: $P_{F_0}(v_o, c_0) = P_{\mathcal{M}_0}(v_o, c_0)$.

## 3.4 Likelihood Maps

In order to learn the above, we assumed access to ground truth likelihood maps used for supervised learning which we obtain as follows. For each grasp in our dataset, instead of an object contact map, we generate a $(2048, N)$ per-gripper-keypoint proximity map of $M$ nearest neighbors (NN) to each of the contact vertices for the canonical keypoints:

$$Prox_o(v_o, k_i) = \begin{cases} 1, & v_o \in NN(\mathcal{V}_G(k_i), M) \\ 0, & \text{otherwise.} \end{cases} \quad (7)$$

where $NN(\mathcal{V}_G(k_i), M) = \{y \in \mathcal{V}_O : |\{z \in \mathcal{V}_O : ||z - \mathcal{V}_G(k_i)|| < ||y - z||\}| < M\}$. We also generate a gripper contact map for the selected keypoints where the contacts are defined as the keypoints closer than a given threshold, to the object point cloud:

$$C_g(k_i) = \begin{cases} 1, & \exists v_o, \; ||\mathcal{V}_O - \mathcal{V}_G(k_i)||^2 < \text{threshold,} \\ 0, & \text{otherwise.} \end{cases} \quad (8)$$

where $Prox_o(v_o, k_i)$ is the object proximity map, and $C_g(k_i)$ is the gripper contact map. For this work, we empirically assumed $M = 20$ and a threshold of 0.04. Finally, the hand-specific object contact map can be obtained as $C_O(v_o, k_i) = Prox_o(v_o, k_i) \cdot C_g(k_i)$.

## 3.5 Grasp Execution at Inference

At test time, the independent unnormalized distribution for $k = 0$, $P_{F_0}(v_o, c_0)$, is leveraged to sample keypoint 0 which will commence the autoregressive inference. Details of this process are provided in Appendix A.2. Moreover, it should be noted that this autoregressive representation does present some limitations. More specifically, the ordering with which the keypoint contacts are being learned and ultimately selected could change the result. However, we refrain from experimenting with all possible combinations of keypoint ordering in the context of this work.

The end-effector joint angles are then inferred by feeding the predicted contact points into an Inverse Kinematics (IK) solver. For our purposes, we used SciPy's Trust Region Reflective algorithm (TRF) [39]. The initial pose given to IK is a heuristic pose calculated by applying a rotation/translation that aligns the palm with the closest object vertex while keeping all non-root joints at their rest pose configuration.

# 4 Experiments

We evaluate our method through the lens of a number of research questions.

**Q1: How successful is the model at producing stable and diverse grasps for various embodiments?** We train our method with a training set containing samples of 5 end-effectors and 38 objects. We then generate grasps on each of the end-effectors and 10 new unseen objects. We evaluate these using the evaluation protocol introduced by [12] which considers 3 out of the 5 end-effectors and tests for grasp stability in Isaac Gym [40]. Similarly to [12], we apply a consistent $0.5ms^{-2}$ acceleration on the object from all $xyz$ directions sequentially, for 1 second each. If the object moves more than $2cm$ after every such application, the grasp is declared as a failure. We also follow the same contact-aware refinement procedure, which applies force closure via a single step of Adam with step size 0.05. In addition, we provide calculated diversity as the standard deviation of the joint angles of all successful grasps, comparably to [12]. There is a limited number of methods that tackle grasping across multiple embodiments. For our comparisons, for the recent methods, we chose GenDexGrasp [12], which assumes hand-agnostic contact maps, AdaGrasp (initOnly as it is the closest setup to our task) [14] which assumes table top grasping only and parameterizes grasps as a pick location and z-axis rotation, and finally DFC [19] which is a differentiable force closure synthesis method. In summary, we selected a set of methods that look at the cross-embodiment grasping problem through various different lenses.

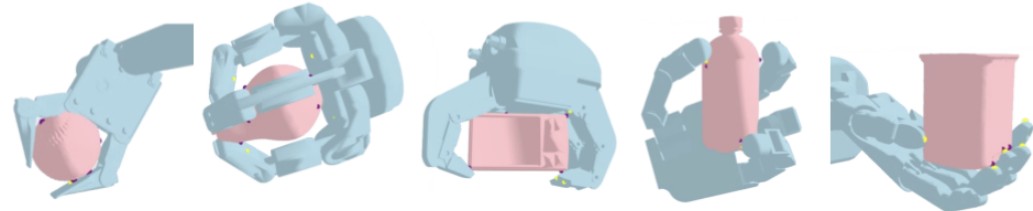

Figure 4: **Qualitative results.** Generated grasps using GeoMatch on **unseen** objects with ezgripper, barrett, robotiq-3finger, allegro and shadowhand. For each grasp, another perspective is included where the GeoMatch predicted keypoints on each object are marked with purple and the gripper user selected keypoints matching these, are marked with yellow.

Results can be found in Tab. 1. In addition, we provide a number of qualitative results in Fig. 4. More qualitative results in the form of rendered grasps can be seen in Fig. 1.

| Method | Success (%) ↑ | | | | Diversity (rad) ↑ | | |
|---|---|---|---|---|---|---|---|
| | ezgripper | barrett | shadowhand | **Mean** | ezgripper | barrett | shadowhand |
| DFC [19] | 58.81 | 85.48 | 72.86 | 72.38 | 0.3095 | 0.3770 | 0.3472 |
| AdaGrasp [14] | 60.0 | 80.0 | - | 70.0 | 0.0003 | 0.0002 | - |
| GenDexGrasp [12] | 43.44 | 71.72 | **77.03** | 64.01 | 0.238 | 0.248 | 0.211 |
| **GeoMatch (Ours)** | **75.0** | **90.0** | 72.5 | **79.17** | 0.188 | 0.249 | 0.205 |

Table 1: **Success and diversity comparisons.** GeoMatch performs more consistently across end-effectors with a varied DoF number while maintaining diversity of grasp configurations. Success rates are provided per end-effector where the mean is calculated across the 3 end-effectors presented, for ease of review.

In our experiments, we observed that GeoMatch is performing slightly worse (-2%) on the 5-finger gripper Shadowhand than the best performing baseline, however performance for the 2-finger and 3-finger grippers **increases by 5-30%** compared to other methods. Diversity remains competitive to other methods. Overall, the minimum performance observed for GeoMatch is significantly higher than baselines and the average performance multi-embodiments beats all baselines we compared against.

**Q2: Is the multi-embodiment model performing better than a model trained on individual embodiments?** We hypothesize that training our method on data containing a variety of end-effectors will result in learning better geometry representations. To investigate this, we train our method on each single embodiment separately by filtering our dataset for each given end-effector. We then compare against the multi-embodiment model. Each of the single end-effector models is trained only on grasp instances of that gripper while the multi-embodiment model is trained on all 5 end-effectors and objects in the training set. The validation set in all cases contains 10 unseen objects. We provide results in Tab. 2. The model trained on multi-embodiment data is performing **20%-35%** better than single end-effector models. This advocates for the value of multi-embodiment grasping policies as opposed to single model policies trained on more data.

| Method | Success (%) ↑ | | | Diversity (rad) ↑ | | |
|---|---|---|---|---|---|---|
| | ezgripper | barrett | shadowhand | ezgripper | barrett | shadowhand |
| Single embodiment | 40.0 | 70.0 | 40.0 | 0.157 | 0.175 | 0.154 |
| Multi embodiment | **75.0** | **90.0** | **72.5** | 0.188 | 0.249 | 0.205 |

Table 2: Comparisons between the Multi-embodiment model and models trained on individual grippers.

**Q3: How robust is the learned model under relaxed assumptions?** While our method demonstrates compelling results, it has been trained on full point clouds. Acknowledging that this is often

a strict assumption, especially when considering real-world environments, we evaluate robustness of the approach under conditions more similar to real-world robotic data. We experiment with grasp generation using: a) noisy point clouds, b) partial point clouds, and c) partial point clouds including noise. For each of these, we perturbed the object point clouds accordingly, and collected grasps using our method **zero-shot**. Success rate was **77.5%**, **66.7%**, and **67.5%** across end-effectors for each type of augmentation respectively. As demonstrated, our method shows reasonable robustness. Experiment details and a breakdown of numbers can be found at Appendix A.

**Q4: How important are various components of the design?** Finally, we investigate the design decisions of our approach and how they affect performance. More specifically, we perform two ablations:

PointNet++ as the encoder of choice instead of GNN. We evaluate our choice towards GNN by swapping out the two GNN encoders with PointNet++[22], a popular encoder architecture for point clouds. Our results show that GNN was indeed a good choice as it performs better than the Point-Net++ ablation, by **10%** averaging across end-effectors. In addition, we empirically observed a 12x slow down when using PointNet++ due to the difference in the number of model parameters, which also makes GNN more light weight and fast. A breakdown per end-effector can be found in Appendix A.

Non-shared weights between keypoint encoders. We hypothesize that a shared encoder among all end-effectors is beneficial for learning features that represent local geometry and this subsequently, informs autoregressive prediction of keypoints. To validate this hypothesis, we conducted an ablation where we separated the end-effector encoder to 6 separate identical encoders, one per keypoint. Our main model with shared weights across all end-effectors and keypoints outperforms the split encoders by **9%**. Further analysis per end-effector can be found in Appendix A.

# 5 Limitations

Our method showcased that grasp learning can benefit from multi-embodiment data in terms of generalization to new objects as well as robustness. Obtaining large amounts of such multi-embodiment grasping data, especially in real world setups can be challenging, time consuming and expensive. However, given that a single embodiment grasping policy was shown to require more data to perform comparably, we argue that spending resources on a multi-embodiment dataset to yield a policy that performs well across a variety of grippers is a better choice. On another note, as our focus was on the notion of cross-embodiment and if a unified grasping policy is achievable, the point clouds used for this work were complete or slightly noisy/partial models. This does not reflect the distribution of noise encountered in real world point clouds derived from depth camera data. Grasps being directed by a small set of keypoints could be viewed as another limitation. This design choice may indeed limit the areas of a gripper that make contact with an object or maybe prevent prediction of power grasps. This could perhaps be mitigated by choosing a larger set of keypoints for better coverage of the end-effector surface. Lastly, our method relies on the robustness of the IK solution. We empirically observed cases where there was a reasonable grasp solution for a set of predicted keypoints, however the chosen IK solution reached the maximum iteration steps and terminated in some suboptimal configuration.

# 6 Conclusion

This work presented a novel multi-embodiment grasping method that leverages GNN to learn powerful geometry features for object and embodiment representation. Our approach demonstrates that a joint encoder trained on multiple embodiments can better embed geometry in a generalizable fashion and ultimately result in higher grasping success rate on unseen objects. The proposed framework also showcased robustness to more realistic point cloud inputs. Diversity of generated grasps remains competitive while producing such diverse grasps is as simple as conditioning with a different high likelihood starting contact point for the first keypoint. Code and models will be released.

**Acknowledgments**

Authors would like to thank Claas Voelcker for valuable feedback and discussion, as well as Silvia Sellán and Alec Jacobson for providing invaluable resources on Blender visualizations. Finally, we'd like to thank reviewers for constructive feedback that resulted in improving our work.

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

# A  Method Details

## A.1  Autoregressive Matching Module Details

The downprojected geometric embeddings are passed to an autoregressive matching module, $\mathcal{M}_i$, consisting of 5 layers, each responsible for predicting the index of the object vertex $c_n$ where keypoint $k_n$ makes contact, given outputs of keypoints $k_0, ..., k_{n-1}$. Each layer $n$ concatenates the embedding of the $n$-th keypoint of the end-effector along with the object embedding. Then, it calculates the *relative* distance map of each object vertex to each of the $n - 1$ object vertices where the previous $n - 1$ keypoints make contact. Note that is done via teacher forcing: instead of using the predictions of each $n - 1$ layer, we use the previous $n - 1$ ground truth contact points. This avoids error propagation during training. The relative distance maps are stacked and concatenated with the object and $n$-th keypoint embeddings. This constitutes the input to an MLP that predicts a binary classification prediction over the object vertices that indicates the predicted $n$-th contact point $c_n$.

## A.2  Keypoint Sampling at Inference Time

As noted above, the independent unnormalized distribution for $k = 0$ is leveraged to sample keypoint 0 which will commence the autoregressive inference. We use the 0-th dimension as a scoring mechanism for sampling high likelihood points where keypoint 0 makes contact. This is then passed into the model as the previous contact of keypoint 1. At inference time, at the $n$-th step, teacher forcing is substituted with passing in the $(n - 1)$ predicted contact vertices. Finally, the end result is a tensor of 6 coordinates of the object graph. As previously mentioned, grasping is a multi-modal distribution and our model should be able to sample from the various modes. In our method, this can be achieved straightforwardly by sampling a variety of starting top-K points for keypoint 0. The intuition behind this is that diverse, yet likely starting points for keypoint 0 will condition subsequent predicted points differently, and ultimately yield different grasp modes. For our experiments, we sampled 4 such top-K points, namely top-0, 20, 50, 100, in order to explore the capacity of our method to generate diverse grasps. A more sophisticated sampling algorithm such as Beam search, could be applied here, however we empirically achieved sufficient diversity through multimodal sampling of keypoint 0.

# B  Additional Experimental Details

## B.1  Robustness Experiments

To investigate the robustness of the learned representations, we conducted a set of experiments where we tested out our trained model with noisy object point clouds, partial point clouds and partial point clouds with noise added as well:

**Noisy point clouds.** For this experiment, we processed the 10 object point clouds of our evaluation set by adding Gaussian noise with standard deviation 0.001 and clipping to a one standard deviation interval, to each of the points. Our evaluation process was then repeated with these as input **zero-shot**, i.e. grasps were generated and evaluated with the same process.

**Partial point clouds.** For this experiment, we emulated a table top scenario where objects placed on a table would be missing the bottom of their surface. To achieve this, for each object point cloud, we defined a z-plane

$$z_{\text{thres}} = \frac{z_{max} - z_{min}}{6},$$

where $z_{min}$, $z_{max}$ are the minimum and maximum z-value found in each object point cloud respectively. We then remove all points with $z < z_{\text{thres}}$ in order to emulate such a table effect. The resulting point clouds are again used **zero-shot** on our model to predict grasps.

**Noisy partial point clouds.** For this experiments, the table top emulating partial point clouds generated for the previous experiment are augmented with Gaussian noise of standard deviation 0.001 and clipping to a one standard deviation interval. Grasping generation occurs again **zero-shot** on our model, and the evaluation process remains the same as all other experiments.

Comparative results for all 3 experiments against noiseless inputs can be viewed in Tab. 3.

| Augmentation | Success (%) ↑ | | | Diversity (rad) ↑ | | |
|---|---|---|---|---|---|---|
| | ezgripper | barrett | shadowhand | ezgripper | barrett | shadowhand |
| noiseless | 72.5 | 90.0 | **75.0** | 0.188 | 0.249 | 0.205 |
| noisy | **75** | **95.0** | 62.5 | 0.183 | 0.245 | 0.196 |
| partial | 67.5 | 67.5 | 65.0 | 0.181 | 0.207 | 0.197 |
| noisy partial | 65 | 75.0 | 62.5 | 0.143 | 0.227 | 0.212 |

Table 3: Comparisons between noiseless, noisy, partial, and noisy partial object point cloud inputs.

We observe that our model generally demonstrates robustness to noise with performance actually increasing in two out of three evaluated end-effectors. Partial point clouds cause the performance to drop as expected, however the model is still performing at a good level at multi-embodiments.

## B.2 PointNet++ Ablation

Our choice of GCN as a geometry encoder is, of course, not the single architectural option available for representing 3D geometry features, with PointNet++ [22] being a popular choice in the literature. In this ablation, we investigate the efficacy of GCN in the multi-embodiment grasping setup compared to PointNet++ by replacing both our GCN object and end-effector encoders with a PointNet++ architecture[*].

| Encoder | Success (%) ↑ | | | Diversity (rad) ↑ | | |
|---|---|---|---|---|---|---|
| | ezgripper | barrett | shadowhand | ezgripper | barrett | shadowhand |
| GCN [30] | **75.0** | **90.0** | **72.5** | 0.188 | 0.249 | 0.205 |
| PointNet++ [22] | **75.0** | 70.0 | 65.0 | 0.154 | 0.223 | 0.151 |

Table 4: Comparison between GCN and PointNet++ encoder choices.

Results in Tab. 4 show that the GCN encoder variant outperforms the PointNet++ one for the 3-finger and 5-finger gripper while performs on par with it for the 2-finger gripper. The GCN variant is also showing higher diversity of grasps for all 3 end-effectors.

## B.3 Non-Shared Weights Ablation

For our main method, we assumed shared weights between the representations used in the autoregressive modules predicting each keypoint contact. However, it is of interest to investigate how performance gets impacted if each autoregressive module is free to influence geometry representations for the keypoint it is responsible for. We thus, disentangled encoding weights for each of the autoregressive modules by passsing in a separate end-effector encoder in each.

| Ablation | Success (%) ↑ | | | Diversity (rad) ↑ | | |
|---|---|---|---|---|---|---|
| | ezgripper | barrett | shadowhand | ezgripper | barrett | shadowhand |
| Shared weights | **75.0** | **90.0** | **72.5** | 0.188 | 0.249 | 0.205 |
| Non-shared weights | 70.0 | 82.5 | 60.0 | 0.165 | 0.259 | 0.163 |

Table 5: Comparison between shared and non-shared weights of the end-effector encoder for autoregressive learning.

The comparison is provided in Tab. 5 and indicate that training end-to-end with a shared end-effector encoder for all keypoint predictions, is still a significantly better performant choice. The shared weights variant performs **5%-12.5% better** among the 3 sample embodiments than the non-shared weights ablation.

---

[*]We used the implementation from `https://github.com/yanx27/Pointnet_Pointnet2_pytorch`

## B.4    Comparison to GraspIt!

Additionally to baselines presented in the main paper, we include a comparison of our method to the simulated annealing planner in GraspIt! [41] for the Barrett end-effector. We produced 4 grasps with the highest quality as per GraspIt!'s quality metric, and further converted them to the format expected by the IsaacGym evaluation protocol we used for our method. Results can be found in Tab. 6 below.

| Method | Success (%) ↑ | Diversity (rad) ↑ |
|--------|---------------|-------------------|
|        | barrett | barrett |
| GraspIt! [41] | 89.99 | 0.00347 |
| GeoMatch (ours) | 90.0 | 0.24900 |

Table 6: Comparison of GeoMatch and GraspIt!'s simulated annealing planner for barrett.

## B.5    Impact of the number of embodiments included at training

In order to better understand the impact of the number of embodiments seen by the unified grasping policy and investigate potential diminishing returns, we trained variants of the policy with an increasing number of end-effectors and compared all on Shadowhand. The results presented in Tab. 7 suggest that indeed training on multi-embodiment data and adding more end-effectors during training increase performance significantly.

| Training set contains | | | | | Success (%) ↑ |
|-----------|---------|---------|---------|------------|---------------|
| ezgripper | barrett | robotiq | allegro | shadowhand | |
|           |         |         |         | ✓ | 40.0 |
|           | ✓       |         |         | ✓ | 47.5 |
| ✓         | ✓       |         |         | ✓ | 55.0 |
| ✓         | ✓       |         | ✓       | ✓ | 55.0 |
| ✓         | ✓       | ✓       | ✓       | ✓ | 72.5 |

Table 7: Comparison between an increasing number of end-effectors seen at training time. Evaluation is performed on an in-distribution end-effector (Shadowhand) but on our unseen set of objects.

## C    Implementation Details

Implementation of all experiments was done using an Adam optimizer with learning rate of 1e-4 for 200 epochs. An assortment of GPU was used, namely RTX3090, V100, T4. Other hyperparameters used were provided in the main paper but for completeness, we include all hyperparameters here. The GNN used had 3 hidden layers of size 256. The output feature size of the GNN encoder was 512. The two parts of the loss were weighed by 0.5 each while the two positive weights used for the two BCE losses were 500 and 200 for the independent distributions and marginals respectively. The dataset used was the subset of MultiDex used by [12] to train the CMap-CVAE model of their approach, which contains 50,802 diverse grasping poses for 5 hands and 58 objects from YCB and ContactDB. The training set contained 38 objects and the validation set the remaining 10. The projection layer was a Linear layer without bias with an output dimension of 64 and each of the MLP autoregressive modules had 3 hidden layers of size 256.

For the IK, SciPy's TRF algorithm was used where each resulting set of predicted keypoints was moved 5mm away from the surface of the object on the direction of the normal in order to form a pre-grasp pose. The initial pose guess provided, was a heuristic calculated by orienting the palm of the gripper to align with the negative of the normal on the object surface at the closest surface point. For evaluation, 4 grasps per object-gripper pair were sampled by selected the top-[0, 20, 50, 100] most likely keypoint 0.

The Isaac Gym based evaluation scripts from [12] were used as is, aside from the one Adam step of force closure where the step size used was 0.05 in order to make the force closure smoother and less abrupt.

