# OpenReview forum: "Geometry Matching for Multi-Embodiment Grasping"
_robot-learning.org/CoRL/2023/Conference — CoRL 2023 Poster_

### Official Review · Reviewer_U6zR · 2023-07-04

**Confidence:** 5
**Originality:** Very Good
**Technical Quality:** Good
**Clarity Of Presentation:** Good
**Impact:** 4

**Recommendation:**

Weak Accept: I recommend accepting the paper, but will not argue for my recommendation if the majority of other reviewers have a different opinion.

**Review:**

### Quality

The quality of this work is high. The authors provide a clear and detailed description of their method, including the mathematical formulation and implementation details. The simulation experiments are well-designed and the results are presented in a clear and concise manner. The authors also provide a thorough analysis of their results, including a comparison to existing methods and an ablation study to evaluate the contribution of different components of their method. Nevertheless, the authors should also compare their method to the simulated annealing planner in GraspIt! as that is still a relevant baseline for dexterous grasping. Also, some real-world experiments on two to three grippers would defiantly improve the quality of this work substantially.

### Clarity

The paper is structured and oftentimes quite well written. Nevertheless, at times, the paper is hard to understand and requires multiple reads to comprehend fully. For instance, many long paragraphs make reading taxing, and quite some grammatical errors are still in the text.

I have the following suggestions for clarity:
- Consider splitting section 2 on dexterous grasping into more paragraphs. In its current state, that section is just a huge one-paragraph text which is very cognitively heavy to read.
- Figure 1 is not referred to in the text.
- Read through and fix some of the grammatical errors. A tool such as Grammarly would probably catch these grammatical errors.

### Originality

The authors present a novel method for multi-embodiment grasping that uses Graph Neural Networks to learn rich geometric representations of objects and end-effectors. While there have been previous works on multi-embodiment grasping, the authors demonstrate that their method outperforms existing methods and provides competitive diversity of grasps. Using Graph Neural Networks to learn geometric representations is also a novel contribution.

The authors do miss some of the earlier works on dexterous grasping, such as [1,2] and on Geometry-Aware Grasping [3,4]. Please add these to your manuscript.

### Significance

The work addresses the challenging problem of data-driven multi-embodiment grasping. While there have been previous works on multi-embodiment grasping, the authors demonstrate that their non-incremental method outperforms existing methods. The method is also sound when the complete model is known, which is quite a big assumption in real-world grasping. The authors do miss some of the earlier works on dexterous grasping, such as [1,2] and on Geometry-Aware Grasping [3,4]. Please add these to your manuscript.

### Limitations

The limitation points out relevant shortcomings with the method. However, it completely ignores that the method is only evaluated in simulation and that nothing can really be said about real-world performance. Also, it should highlight that the method requires a complete object model or at least one that is close to complete.

### Strengths

- Clear and detailed description of the method
- Well-designed experiments with a thorough analysis of the results
- Novel contribution of using Graph Neural Networks to learn joint object and gripper geometric representations
- Outperforms existing methods and provides competitive diversity of grasps

### Weaknesses

- The method may not generalize well to real-world scenarios with more complex objects and end-effectors
- The keypoints ensure that only those parts of the gipper are in contact with the object, which might prevent generating specific multi-contact grasps such as power grasps.
- The method should also be compared against the non-learning-based simulated annealing method in GraspIt!
- No real-world experiments. I understand that it is unlikely that the authors have all the grippers used in the simulation. Thus, for me, it would be enough just to use two to three grippers.

[1] Varley, Jacob, et al. "Generating multi-fingered robotic grasps via deep learning." _2015 IEEE/RSJ international conference on intelligent robots and systems (IROS)_. IEEE, 2015.

[2] Lundell, Jens, et al. "Multi-fingan: Generative coarse-to-fine sampling of multi-finger grasps." _2021 IEEE International Conference on Robotics and Automation (ICRA)_. IEEE, 2021.

[3] Varley, Jacob, et al. "Shape completion enabled robotic grasping." _2017 IEEE/RSJ international conference on intelligent robots and systems (IROS)_. IEEE, 2017.

[4] Lundell, Jens, Francesco Verdoja, and Ville Kyrki. "Robust grasp planning over uncertain shape completions." _2019 IEEE/RSJ International Conference on Intelligent Robots and Systems (IROS)_. IEEE, 2019.

**Quality Of The Limitations Section:**

Additional details required

**Questions For Rebuttal:**

- What is $M_i$ in Equation 2?
- I have problems understanding the meaning of Equation 3 and the loss in Equation 4. I understood the loss in Equation 6, but I do not understand why you need the loss in Equation 4.
- On lines 184-185, what is mean by: "Each layer n concatenates the embedding of the n-th keypoint of the end-effector along with the object embedding"
- It would be interesting to see some results of Q2 in Section 4 but with models trained on 1, 2, 3, 4, and 5 end-effectors respectively. How much the success rates improve with the number of end-effectors could perhaps indicate if there are diminishing returns the number of grippers to be used.
- Could you do the experiments but with point-clouds that remove self occluded regions? That would be the closest to a real-world experiment you could probably come.

**Robotics Focus:**

Relevant but unlikely to deploy to hardware in near future

**Summary Of Paper:**

This paper addresses a timely problem in data-driven dexterous grasping: how to train a single grasp sampler that generalizes over gripper morphologies? To solve this problem, a graph neural network named GeoMatch is introduced to learn rich geometric representations for both objects and end-effectors. GeoMatch takes as input both the object and the gripper morphologies as graphs and outputs, autoregressively, the contact region on the object for one finger at a time. The final gripper configuration can then be extracted from the contact regions using an inverse kinematics algorithm. The experimental evaluation in simulation shows that GeoMatch outperforms three other learning-based methods that do not explicitly model both objects and end-effectors geometries jointly.

**Summary Of Recommendation:**

The proposed GeoMatch is a data-driven method to enable multi-embodiment grasping. The method is novel and non-incremental and the results are good with better grasp success rates than the baseline. The lack of comparison to GraspIt!'s simulated annealing planner and no real-world experimental evaluation is the main limitations. Nevertheless, this reviewer recommends a weak acceptance due to the novelty of the work, but if the paper would include a comparison to GraspIt! and some real-world experiments, then it would most probably be a strong acceptance.

---

> ### Author Response · Authors · 2023-08-08
> **Response to Reviewer U6zR**
>
> Thank you to the reviewer for their time and feedback. Please see our responses below:
>
> __Splitting section 2 on dexterous grasping into more paragraphs for clarity__
>
> We are happy to update section 2 accordingly, thank you for the suggestion.
>
> __Figure 1 not referenced__
>
> Thank you for pointing this out, it should have been referenced under qualitative results. We will fix this.
>
> __Grammatical errors__
>
> Thank you for highlighting these errors. We will address them in the next revision.
>
> __Missing references__
>
> Thank you for pointing out missing references. We will add these to the paper.
>
> __Limitation of method requiring a complete object model or at least one that is close to complete__
>
> The method indeed has not yet been tested with object point clouds obtained by real cameras. We only did a limited robustness experiment via noisy and partial point clouds as our main focus was the concept of cross-embodiment for grasping. We will highlight this more in our limitations section as: "As our focus was on the notion of cross-embodiment and if a unified grasping policy is achievable, the point clouds used for this work were complete or slightly noisy/partial models . However, it should be highlighted that point clouds generated from real camera depth information have a different noise distribution".
>
> __Keypoints limiting which parts of the gripper attach__
>
> We will discuss this in the limitations section and update the manuscript.
>
> __Comparison to GraspIt!__
>
> We will try to add GraspIt! comparisons in the manuscript by end of rebuttals.
>
> __Lack of real-world experiments__
>
> Quantitative experiments at scale, especially for cross-embodiment, are often done in SIM, e.g. GenDexGrasp [11], Fast-Grasp'D [18], however we are working on setting up a demo in real using Allegro on a Franka arm by camera ready if accepted.
>
> __What is Mi.__
>
> Mi represents the autoregressive modules, each responsible for predicting the i-th keypoint contact. We will clarify this in the manuscript.
>
> __Meaning of Equation 3 and the loss in Equation 4__
>
> Equation 3 is the dot product between the object and end-effector keypoint embeddings and is optimized with a BCE (Equation 4) loss against the object contact map tiled by the keypoint number. For each keypoint, the corresponding object contact map copy has 1s at the areas where that keypoint attaches. If a keypoint does not attach anywhere for a given sample, it is all 0s. Basically this serves two purposes: a) regularizer so that the overall contact maps predicted per keypoint do not veer too far from seen contact maps since these are definitely valid, and b) since keypoint 0 at inference time is a free degree of freedom, the probability map of keypoint 0 can guide choosing a high likelihood keypoint 0 (compared to what has been seen in the training set as valid grasps) that will not lead the remaining autoregressive prediction completely out of distribution.
>
> __Explanation of ln. 185__
>
> Each of the n layers takes as input the object embedding concatenated with the gripper embedding but only on the n-th keypoint as opposed to the full gripper embedding since the n-th keypoint is the one that the n-th layer should pay attention to. In other words, let's say the full end-effector embedding Eg for all keypoints is of size (bs x 6 x 256) where 256 is the embedding size and 6 is the number of keypoints, what is concatenated with the object embedding is Eg[:, n, :]. We will integrate this in the manuscript to make this clearer for the reader.
>
> __Experiment on increasing number of end-effectors__
>
> This is actually a really cool experiment to run! Please see numbers below. Each model has been trained with the end-effectors indicated, while all are evaluated with Shadowhand only and an unseen object eval set (i.e. objects at eval time are not in the training set). Order of including end-effectors was selected randomly.
>
> | Ezgripper  | Barrett | Robotiq   |   Allegro | Shadow | Success (%) |
> | :---             |    :----:  |       :----:  |       :----:  |      :----:  |                ---: |
> |                   |             |               |                |      x       |         40.0      |
> |                   |      x     |               |                |      x       |         47.5      |
> |         x        |      x     |               |                |      x       |         55.0      |
> |         x        |      x     |               |       x       |      x       |         55.0      |
> |         x        |      x     |       x      |       x       |      x       |         72.5      |
>
> __Experiments removing self occluded regions__
>
> We agree this is an interesting experiment. The scope of this work was to evaluate the concept of cross-embodiment and learning of a unified grasping policy thus we did not focus too much on its instantiation under imperfect sensing, however providing a demo in real for camera ready if accepted, will showcase application on more realistic point clouds.

---

> > ### Comment · Reviewer_U6zR · 2023-08-09
> > **Response to rebuttal**
> >
> > Thanks for the clarification, and good work with the rebuttal. Your response adequately answers most of the questions I had.
> >
> > The new experiment on increasing the number of end-effectors is a very valuable contribution. There seems to be a trend that the more grippers trained on the higher the success rate, which is a reasonable result. Do you have any hypothesis why adding the Allegro hand does not increase the success rate but including the Robotiq does? Robotiq seems to be very similar to the Barrett hand, while the Allegro hand is quite different from the other grippers.
> >
> > Overall, my recommendation still holds at a weak accept as real-world experiments, and a comparison to another analytical method such as GraspIt! is still missing.

---

> > > ### Author Response · Authors · 2023-08-15
> > > **Response to Reviewer U6zR**
> > >
> > > Thank you for the response, we're glad to hear the experiment was valuable. Our hypothesis is that while Allegro has a different number of fingers than all the others, the way its grasps look do present an overlap with Shadowhand due to their morphologies. E.g. some Allegro grasps could be seen as degenerate Shadowhand grasps that do not have all five fingers making contact. In contrast, while both Barrett and Robotiq have 3 fingers, their fingers' position with respect to palm as well as the length of their fingers differ enough that their respective grasp distributions are adequately different.
> > >
> > > We would also like to report evaluation numbers of GraspIt! for __Barrett__ on our unseen set of objects. GraspIt! was used for grasp generation. We then used the same IsaacGym evaluation protocol to assess success.
> > >
> > > |                    | Success (%) |   Diversity |
> > > | --------------- | ------------------|  -------------|
> > > |  GraspIt!     |        89.99      |   0.00347  |
> > > | GeoMatch   |       90.00      |   0.24900  |
> > >
> > > Finally, we got a preliminary demo on a real robot with Allegro working yesterday evening. Please see https://geo-match.github.io/ for the video. It should be noted that while this grasp was performed with an Allegro hand, technically this is a __OOD__ gripper since this is a right Allegro hand as opposed to the left Allegro that was used in training. We will work on adding more real demos for camera ready, if accepted.

---

### Official Review · Reviewer_vXSr · 2023-07-20

**Confidence:** 3
**Originality:** Very Good
**Technical Quality:** Good
**Clarity Of Presentation:** Good
**Impact:** 3

**Recommendation:**

Weak Accept: I recommend accepting the paper, but will not argue for my recommendation if the majority of other reviewers have a different opinion.

**Review:**

Strengths:
1. The paper addresses an essential issue in dexterous grasping. The suggested method is novel and demonstrates impressive experimental results.
2. The concept of an autoregressive model to predict the contact of each keypoint sequentially is appealing.

Weaknesses:
1. While the paper emphasizes the need for transferability to new end-effector geometries, the experiments do not demonstrate the model's ability to generalize to new end-effectors.
2. Although the paper is generally understandable, the method section seems a bit wordy.


**Quality Of The Limitations Section:**

Limitations are addressed clearly

**Questions For Rebuttal:**

1. Fig. 3b is not referenced or explained in the paper. What does the minus sign in the figure represent?
2. In Section 3.4, the authors state `More specifically, the ordering with which the keypoint contacts are being learning and ultimately selected could vastly change the result. However, we refrain from experimenting with all possible combinations of keypoint ordering in the scope of this work.` However, the method of `experimenting with all possible combinations of keypoint` is unclear. Do you train the model with all possible combinations?
3. Following the previous point, it it is not clear how the sequence of the keypoints are decided. For example, will the finger keypoints be estimated before the palm, or vice versa?
4. Why does Table 1 only show the result for three grippers instead of all five? Is the `Mean` averaged over five grippers?
5. There are some typo in the paper. It would be ideal to do a proof-reading or run the paper through a grammar checker.
    1. Line 126, `It is recommended to selected keypoints having …` -> ‘It is recommended to select keypoints having …’
    2. Line 181, there is a comma after `Both`.
    3. Line 231, `being learning` should be ‘being learned’.


**Robotics Focus:**

Highly relevant to robotics but no hardware experiments

**Summary Of Paper:**

The paper studies multi-embodiment in dexterous grasping. It presents a method based on a Graph Neural Network (GNN) that estimates the contact likelihood for each of the keypoints in a high-degree-of-freedom end-effector. This facilitates the prediction of grasps across various end-effector models. Experimental evaluations show that the proposed method surpasses a number of baselines in terms of grasp success rate in simulations.

**Summary Of Recommendation:**

The paper offers a novel method for dexterous grasping that can generalize across different gripper geometries. I believe the paper contributes valuable insight to the field, thus I recommend accepting the paper. However, I believe it is essential to demonstrate the generalization ability of the proposed method to novel gripper geometry, which will make the paper much more convincing.

---

> ### Author Response · Authors · 2023-08-08
> **Response to Reviewer vXSr**
>
> Thank you to the reviewer for their time and feedback. Please see our responses below:
>
> __Generalization to OOD grippers not shown__
>
> Our work focuses on how cross-embodiment benefits performance across end-effectors compared to only working with single-gripper policies. We do, however, anticipate that this method would generalize better to unseen grippers had it seen a larger variety of them during training.
>
> __Fig. 3b is not referenced__
>
> While explained in the text, it is not explicitly referenced so that the reader can make the connection, you are correct. We will fix this.
>
> __Section 3.4: Do you train the model with all possible combinations of keypoints?__
>
> We only train on the order we used to select keypoints per end-effector at the beginning. This order is chosen once.  A different initial selection order could result in some variation of the results although we wouldn't expect much as the order was random. When we selected keypoints, we mostly did so either left to right for ezgripper or fingers first, palm last for all others but there was no specific reason other than that it felt more natural. One could choose differently. We will update section 3.4 to make this clearer.
>
> __Tables presenting 3 out of 5 end-effectors in results.__
>
> We will clarify this in the paper as well. The mean is taken over the three reported  end-effectors.  To allow for fair baseline comparison, we followed the evaluation protocol provided by GenDexGrasp [12], which only considers 3 of the 5 end-effectors.
>
> __Typos__
>
> Thank you so much for pointing these out. We will carefully proofread the text as we update the draft.

---

> > ### Comment · Reviewer_vXSr · 2023-08-14
> > **Response to authors**
> >
> > The reviewer thanks the authors for their rebuttal. Most of my concerns are addressed, while I think demonstrating generalization to OOD grippers will significantly strengthen the paper (as it is one of the motivation of the paper). I keep my recommendation for accepting the paper.

---

> > > ### Author Response · Authors · 2023-08-15
> > > **Response to Reviewer vXSr**
> > >
> > > Thank you for the response.
> > >
> > > While this is not a full evaluation of OOD grippers, we were able to demonstrate some OOD behavior on a real world robot through a preliminary demo. Even though this grasp was performed with an Allegro hand, technically this is a OOD gripper since this is a right Allegro hand as opposed to the left Allegro that was used in training. We will work on adding more real demos for camera ready, if accepted.
> > > Please see https://geo-match.github.io/ for the video.

---

### Official Review · Reviewer_dMoq · 2023-07-20

**Confidence:** 5
**Originality:** Good
**Technical Quality:** Very Good
**Clarity Of Presentation:** Very Good
**Impact:** 3

**Recommendation:**

Weak Accept: I recommend accepting the paper, but will not argue for my recommendation if the majority of other reviewers have a different opinion.

**Review:**

The paper models the grasp prediction problem for multi-embodiment (gripper model) scenarios by focusing on the likely object contact points given the gripper model. In the discussion of related work, the authors bring out the current research gap, and how they seek to address some parts of it. I feel that the most close work to the overall idea is UniGrasp [13] as it also relies on a sequential point prediction conditioned upon previous predictions. The proposed method based on GNN of selected gripper points is able to scale well to high-DoF grippers and it looks to be a simple but effective design choice. The intuition for each part is brought out clearly and writing/figures seem clear from the context.

I don't believe that this is exactly a solved problem (nor do the authors claim so) and hence readers would like some more insight in the limitations section about what parts of the entire pipeline are the bottleneck in terms of giving good results. Overall, this is a good extension of existing works tackling the problem and seems to be an effective method and modeling design to tackle highly different geometries of grippers.



Strengths:

- Clear motivation for the probelem and design choice for the modeling strategy. Object centric viewpoint is a natural way to keep the procedure somewhat gripper agnostic. Key related works are sufficiently covered.

- The key limitations of prior works seem to be sufficiently handled in terms of scaling to high-DoF grippers and unseen objects. The gripper encoding scheme and use of GNN instead of PointNet++ helps to bridge prior limitations.

- The authors perform adequete ablation studies for their design choices and the proposed method shows strong improvement when compared to existing related work. Results from simulation seem promising and such a method can perhaps future works tackling a generalized, multi-emboidment settings.


Weakeness:

- The proposed way to handle the different geometries does not really explicitly (or even implicitly?) encode the gripper kinematics. Although simply using the geometry information (via the canonical gripper point cloud) still shows good results, the contribution then is a bit limited as the gripper encoding is simply a somewhat manual process.

- Lack of real world experiments and inference runtime discussion? On average how many grasp candidates have a valid I.K solution? So as whole, how does the proposed pipeline's runtime look like?

- In RQ-2 about comparing to single embodiment focused models, the ideal comparison should be to methods which target a single gripper and whose output is a pose + joint angle vector since those are some strong baselines to compare against.

- The experiment with partial point cloud assumes that only the bottom part of the object is not visible with the point cloud. But this is a strong assumption to make as in real world settings, due to the camera pose, major parts of the object can be missing. Its not a strong limitation as the paper does not claim to work with partial point clouds. The experiment results show a high performance with partial inputs which is not strictly correct in some sense. Maybe a clarifying remark in the main paper will help readers (instead of appendix).


**Quality Of The Limitations Section:**

Additional details required

**Questions For Rebuttal:**

- Does the proposed way to encode the gripper geometry also somehow handles the gripper kinematics (i.e how the joints will move). Is there any intuition as to how the model resolves this or adapts to this ambiguity just from the geometry + training data?

- Some details about the runtime for the model, even if the numbers are from simulation. How many proposals actually pass the I.K stage on average?

- How is the 6D gripper pose inferred from the predicted contacts? Are the contacts predicted in a pre-defined order (with some correspondence to gripper fingers?)

**Robotics Focus:**

Highly relevant to robotics but no hardware experiments

**Summary Of Paper:**

This paper focuses on the problem of generalized grasp synthesis across different objects and gripper models. It addresses the research gap of having a single, general method for grasping that works with different grippers without requiring re-training for each specific gripper. The method utilizes geometry information from the object and gripper point clouds in GNN setting	and predicts contact likelihood. This works along with a sequential, auto-regressive prediction for 6 keypoints corresponding to the gripper palm and a maximum of 5 fingers. Once the correspondence on the object points is established, the grasp is recovered using an existing I.K solver. Diverse grasps are generated by considering top-K likely points for initial contact and subsequent iterations. Experiments are conducted entirely in simulation and show promising performance w.r.t existing baselines.

**Summary Of Recommendation:**

The proposed work focused on the resolving the gap left by prior works (UniGrasp [13]) in terms of predicting contact points for high-DoF and > 3 finger grippers. The method is able to scale well to such configurations and shows promising results in simulation, although the real world experiments and more discussion on probable limitations/pitfalls could help the readers since some assumptions might make it hard to deploy to real world use case!

---

> ### Author Response · Authors · 2023-08-08
> **Response to Reviewer dMoq**
>
> Thank you to the reviewer for their time and feedback. Please see our responses below:
>
> __Does the proposed method encode gripper kinematics explicitly or implicitly__
>
> A canonical pose was chosen as input to the encoder arbitrarily. The only condition placed on it was that keypoints are visible i.e. the end-effector is not deformed. In practice, we set each joint to the mean of its angle limits. However, training optimizes learning of contact points based on valid grasps available in the dataset. The training data contains multi-modal grasps for each of the end-effectors and objects. We hypothesize that optimizing towards learning a distribution of how vertices on the object match with contact points on the end-effectors implicitly encode the gripper's kinematics. In other words, learning where a gripper's fingers and palm "land" on an object implicitly encodes how this gripper is able to move.
>
> __On average how many grasp candidates have a valid I.K solution?__
>
> We seed the gradient-descent-based I.K solution with a valid pose, therefore at termination it is guaranteed to return a valid pose with joints within their joint limits. We also do not check the residual of the I.K output against the desired target pose (inferred by the network), as doing so has very little impact on performance. This step can be seen as projecting the model pose onto the space of valid poses using a model-based IK.
>
> __Discussion about inference runtime for forward pass and IK.__
>
> Producing contact keypoints for a grasp (aka forward pass of the model) is <=0.1 seconds on CPU. The IK we used was not optimized and can take up to a few minutes per sample for some end effectors. However, we should note that we did not focus on grasp generation speed as part of this work. There are many off-the-self fast and stable IK solutions available that can perform IK significantly faster (e.g. the IK in pybullet runs in less than 20ms for a Kuka iwwa7 arm with a parallel jaw end effector).
>
> __Ideal comparison should be to methods which target a single gripper and whose output is a pose + joint angle vector__
>
> Thank you for the suggestion. Do you have a specific baseline in mind that you would like to see in a comparison?
>
> __Partial point clouds not being too realistic in real scenarios.__
>
> We are happy to add a clarifying remark to this effect in the main paper. Perhaps adding the following in the limitation section helps clarify: "As our focus was on the notion of cross-embodiment and if a unified grasping policy is achievable, the point clouds used for this work were complete or slightly noisy/partial models . However, it should be highlighted that point clouds generated from real camera depth information have a different noise distribution". Please also note that we are working on setting up a demo in real using Allegro on a Franka arm by camera ready if accepted, which will demonstrate application on real world models.
>
> __Are the contacts predicted in a predefined order?__
>
> Contacts are predicted indeed in a pre-defined order due to the autoregressive nature of the method. This coincides with the order in which the one time user-selected keypoints of interest were chosen at the beginning. A different initial selection order could result in some variation of the results although we wouldn't expect much as the order was random. When we selected keypoints, we mostly did so either left to right for ezgripper or fingers first, palm last for all others but there was no specific reason other than that it felt more natural. We do briefly discuss this in Sec. 3.4, but we can reword and expand this a bit to make it clearer.

---

> > ### Comment · Reviewer_dMoq · 2023-08-11
> > **Response to rebuttal**
> >
> > Thank you for your prompt reply to my questions and clarifying certain aspects about the paper.
> >
> > I think the comment about gripper kinematics impcitly being learned via the learning of gripper-object contact points is quite interesting. Perhaps adding this insight to paper/appendix would be helpful to readers!
> >
> > For a baseline against methods that directly predict the pose + joints, you could consider works like FFHNet [1], DVGG [2] or similar. I agree that a joint model for multi-embodiment grasping is attractive, however its true usefulness will be if its competitive with works that just target one embodiment. I don't expect it to be better than a significant margin but some level of competititveness can motivate the contribution even more. Even a simple comparsion with any work targeting a gripper from your paper's subset can an effective experiment to carry out (not a requirement for the rebuttal).
> >
> > I agree that a real-time algorithm need not be the main focus of the work, but then a comparison with some slower, analytical methods like Graspit (as suggested by Reviewer-U6zR) could also be helpful as they also perform a contact based optimization using the object's known shape. I do like the fact about trying to validate on GenDexGrasp's simulation setup since that is perhaps the most close work to yours. I also feel the lack of any real world demo/experiments also keeps my recommendation at a weak accept even though its understood that most related works often just validate in simulation.
> >
> > **References**
> > 1. Mayer, Vincent, et al. "Ffhnet: Generating multi-fingered robotic grasps for unknown objects in real-time." 2022 International Conference on Robotics and Automation (ICRA). IEEE, 2022.
> > 2. Wei, Wei, et al. "DVGG: Deep variational grasp generation for dextrous manipulation." IEEE Robotics and Automation Letters 7.2 (2022): 1659-1666.

---

> > > ### Author Response · Authors · 2023-08-15
> > > **Response to Reviewer dMoq**
> > >
> > > Thank you for the response and for suggesting suitable baselines. We will try to incorporate them for camera ready, if accepted.
> > >
> > > As responded to Reviewer U6zR, we were able to include success evaluation of GraspIt! on Barrett as well as a preliminary demo on a real robot. We quote our response here for convenience:
> > >
> > > "We would also like to report evaluation numbers of GraspIt! for __Barrett__ on our unseen set of objects. GraspIt! was used for grasp generation. We then used the same IsaacGym evaluation protocol to assess success.
> > >
> > > |                    | Success (%) |   Diversity |
> > > | --------------- | ------------------|  -------------|
> > > |  GraspIt!     |        89.99      |   0.00347  |
> > > | GeoMatch   |       90.00      |   0.24900  |
> > >
> > > Finally, we got a preliminary demo on a real robot with Allegro working yesterday evening. Please see https://geo-match.github.io/ for the video. It should be noted that while this grasp was performed with an Allegro hand, technically this is a __OOD__ gripper since this is a right Allegro hand as opposed to the left Allegro that was used in training. We will work on adding more real demos for camera ready, if accepted."

---

### Official Review · Reviewer_AGep · 2023-07-21

**Confidence:** 3
**Originality:** Fair
**Technical Quality:** Good
**Clarity Of Presentation:** Good
**Impact:** 3

**Recommendation:**

Weak Accept: I recommend accepting the paper, but will not argue for my recommendation if the majority of other reviewers have a different opinion.

**Review:**

Strength:
- The authors have provided a comprehensive literature review on grasping and graph neural networks, along with a discussion on these existing works. This paper describes an end-to-end learning framework that proposes grasp points for various end-effectors and the trained model generalizes to unseen objects. It presents simulated experiments that demonstrate how the proposed method improves the robustness of generated grasps against disturbances and enhances grasp diversity compared to three baseline methods: DFC, AdaGrasp, and GenDexGrasp.

Weakness:
- The presentation of Section 3 could benefit from improvements in clarity and readability. While this section provides abundant technical details, the connections and explanations between them could be made clearer. A possible improvement could be incorporating an algorithmic presentation, which would provide a step-by-step illustration of the processes discussed. Additionally, augmenting or supplementing Figure 3 with a larger, more detailed visualization could help in conveying the information more effectively. This would allow readers to better comprehend the intricate technical details and the connections among them.
- Section 4 could use a more thorough discussion concerning the selection of the baseline methods. It's important to justify why these specific methods were chosen and how they technically compare to the proposed method. The authors should highlight the unique characteristics of each selected method and elucidate why they are relevant for a side-by-side evaluation. This could include discussing the similarities and differences in methodology, the principles they are based upon, or how they approach the problem differently. By providing such context, readers will gain a better understanding of the strengths and weaknesses of the proposed method in comparison to established techniques in the field.
- While the term "dexterous grasp" is frequently mentioned throughout the article, the experimental evaluation focuses predominantly on a simpler grasping task involving resistance to disturbances. The learned geometry representations can be useful for most manipulation tasks. It would be considerably more compelling to see how the proposed method influences more complex tasks. Activities such as object reorientation or object transfer, which demand a higher degree of dexterity and precision from the robotic end-effectors, would provide a more rigorous testing ground for the method.
- The current experimental results focus on evaluating the stability and diversity of grasps for various robotic grippers in a simulated environment. However, it would significantly enhance the relevance of this work to the community if the authors could demonstrate how effectively the learned model transfers to real robot grippers. Even a demonstration with just two different types of robot grippers would provide valuable insights into the practical utility of the proposed method. The experiment in Q3 (line 273) simulates only perception noise, neglecting many other physical and control uncertainties present in real-world robotic systems.

**Quality Of The Limitations Section:**

Limitations are addressed clearly

**Questions For Rebuttal:**

- Would the proposed method generalize to unseen end-effectors if the training dataset is generated using a larger variety of grippers?
- Line 243 mentions that five end-effectors were used in the evaluation. Is there a specific reason why Tables 1 and 2 only demonstrate the data for three end-effectors?

**Robotics Focus:**

Highly relevant to robotics but no hardware experiments

**Summary Of Paper:**

The authors propose equipping robotic agents with a sense of geometry via a generalized geometry embedding. This embedding represents both objects and end-effectors, helping predict stable grasps. They propose using Graph Neural Networks (GNN) to encode meaningful geometry representations and use them to predict keypoint contacts on the object surface in an autoregressive manner. The authors' contributions include formulating robot grasp learning as a geometry matching problem, introducing a novel method named GeoMatch that learns expressive geometric embeddings, and comparing GeoMatch against baselines across multiple embodiments.

**Summary Of Recommendation:**

The reviewer believes that the proposed method has considerable potential for enhancing multi-embodiment grasping and even dexterous manipulation. However, the current state of the paper could be improved with a more coherent presentation in Section 3 and a more comprehensive experimental section.


-----------------------------------
Post-Rebuttal:
I appreciate the authors' response to the comments. The clarity of the manuscript has improved, so I have raised my rating to a weak accept.

---

> ### Author Response · Authors · 2023-08-08
> **Response to Reviewer AGep**
>
> Thank you to the reviewer for their time and feedback. Please see our responses below:
>
> __Clarity and readability issues in Section 3__
>
> We will be updating this section along with enhancing Figure 3 to better convey how all the pieces fit together. We will upload a new
> manuscript for your review in a few days.
>
> __Need to discuss selection of baselines in Section 4__
>
> There is a limited number of methods that tackle grasping across multiple embodiments. Of the recent methods, we chose
> GenDexGrasp [12], which assumes hand-agnostic contact maps, AdaGrasp [14] which assumes table top grasping only and
> parameterizes grasps as a pick location and z-axis rotation, and finally DFC [17] which is a differentiable force closure synthesis
> method. In other words, we selected a set of methods that look at the cross-embodiment grasping problems through various different
> lenses. These baseline discussion details will be included in the new manuscript we will upload for your review.
>
> __The term 'dexterous grasp' implying more complex grasping tasks than just grasp generation__
>
> Indeed the term "dexterous" is overloaded and not well defined in the literature. We primarily used the term in line with GenDexGrasp
> [12], Fast-Grasp'D [18], etc. In this work, we focused more on grasping stability and diversity, therefore object reorientation or object
> transfer are out of scope, but are extensions for future work.
>
> __Lack of real experiments__
>
> Quantitative experiments at scale, especially for cross-embodiment, are often done in SIM, e.g. GenDexGrasp [11], Fast-Grasp'D
> [18], however we are setting up a demo in real using Allegro on a Franka arm for camera ready if accepted.
>
> __Would the proposed method generalize to unseen end-effectors if the training dataset is generated using a larger variety of grippers__
>
> We have compared our method using a variable number of end-effectors and have found that there is a clear trend that increasing
> the number of training set end-effectors improves performance. We believe that with more end-effectors the performance will
> continue to improve. Please see results of this experiment in our response to reviewer U6zR.
>
> __Tables presenting 3 out of 5 end-effectors in results.__
>
> In the interest of baseline comparison, we followed the evaluation protocol provided by GenDexGrasp [12], which only considers 3 of
> the 5 end-effectors.

---

### Decision · Program_Chairs · 2023-08-30

**Decision:**

Accept (Poster)

**Comment:**

The authors suggest enhancing robots' grasp capabilities using generalized geometry embeddings, which predict stable grasps by representing objects and end-effectors. They propose Graph Neural Networks (GNN) for encoding meaningful geometry representations to predict keypoint contacts on objects in an autoregressive manner. Their contributions involve framing grasp learning as a geometry matching task, introducing the GeoMatch method for learning rich geometric embeddings, and evaluating GeoMatch against baselines in various scenarios in Isaac Gym. **After rebuttal**: The authors have managed to demo on a real robot demo with Allegro and baselines on GraspIt!, so it's encouraged to add this demo to the final version.

**Strengths**:

- A comprehensive literature review on grasping and graph neural networks, along with a discussion on these existing works

- An end-to-end learning framework that proposes grasp points for various end-effectors and the trained model generalizes to unseen objects

- Adequate ablation studies for their design choices and the proposed method shows strong improvement when compared to existing related work

**Weaknesses**:

- The experiment with partial point cloud assumptions is not too realistic

- While the paper emphasizes the need for transferability to new end-effector geometries, the experiments do not demonstrate the model's ability to generalize to new end-effectors.

Post-rebuttal: Please revise the paper in line with the rebuttal discussion for the camera-ready submission.